# The Impact of Viral and Bacterial Co-Infections and Home Antibiotic Treatment in SARS-CoV-2 Hospitalized Patients at the Policlinico Tor Vergata Hospital, Rome, Italy

**DOI:** 10.3390/antibiotics12091348

**Published:** 2023-08-22

**Authors:** Andrea Di Lorenzo, Laura Campogiani, Marco Iannetta, Roberta Iannazzo, Alessandra Imeneo, Grazia Alessio, Veronica D’Aquila, Barbara Massa, Ilenia Fato, Lorenzo Vittorio Rindi, Vincenzo Malagnino, Elisabetta Teti, Massimo Andreoni, Loredana Sarmati

**Affiliations:** 1Department of System Medicine, Tor Vergata University, 00133 Rome, Italy; andrea.dilorenzo@ptvonline.it (A.D.L.); laura.campogiani@ptvonline.it (L.C.); marco.iannetta@uniroma2.it (M.I.); roberta.iannazzo@ptvonline.it (R.I.); alessandra.imeneo@ptvonline.it (A.I.); grazia.alessio@ptvonline.it (G.A.); veronica.daquila@ptvonline.it (V.D.); barbara.massa@ptvonline.it (B.M.); ilenia.fato@ptvonline.it (I.F.); l.rindi@gmail.com (L.V.R.); vincenzo.malagnino@uniroma2.it (V.M.); elisabetta.teti@ptvonline.it (E.T.); andreoni@uniroma2.it (M.A.); 2Infectious Disease Clinic, Policlinico Tor Vergata, 00133 Rome, Italy

**Keywords:** COVID-19, SARS-CoV-2, co-infection, antibiotics

## Abstract

Co-infections during COVID-19 may worsen patients’ outcomes. This study reports the results of a screening assessing the presence of co-infections among patients hospitalized for SARS-CoV-2 infection in the Infectious Diseases-Ward of the Policlinico Tor Vergata Hospital, Rome, Italy, from 1 January to 31 December 2021. Data on hepatitis B and C virus, urinary antigens for legionella pneumophila and streptococcus pneumoniae, pharyngeal swab for respiratory viruses, QuantiFERON^®^-TB Gold Plus assay (QFT-P), blood cultures and pre-hospitalization antibiotic prescription were recorded. A total of 482 patients were included, 61% males, median age of 65 years (IQR 52–77), median Charlson comorbidity index of 4 (IQR 2–5). The mortality rate was 12.4%; 366 patients needed oxygen supply. In total, 151 patients (31.3%) received home antibiotics without any association with the outcome. No significant association between mortality and the positivity of viral hepatitis markers was found. Out of 442 patients, 125 had an indeterminate QFT-P, associated with increased mortality. SARS-CoV-2 was the only respiratory virus detected among 389 pharyngeal swabs; 15/428 patients were positive for *S. pneumoniae*; none for *L. pneumophila*. In total, 237 blood cultures were drawn within 48 h from hospital admission: 28 were positive and associated with increased mortality. In our cohort, bacterial and viral co-infections in COVID-19 hospitalized patients were rare and not associated with higher mortality.

## 1. Introduction

A new *betacoronavirus* was first recognized in Wuhan, China, in December 2019 and rapidly became a pandemic [1]. The virus was named Severe Acute Respiratory Syndrome Coronavirus 2 (SARS-CoV-2), and the associated disease was defined coronavirus disease 2019 (COVID-19) [2]. The clinical manifestation of SARS-CoV-2 may vary from asymptomatic infection to severe respiratory disease, requiring hospitalization in an intensive care unit (ICU) [3].

It is known that viral respiratory infections, such as influenza, predispose to secondary bacterial and fungal infections, increasing morbidity and mortality [4,5,6,7]. Indeed, bacterial co-infections can complicate COVID-19 [8,9,10,11,12,13]. Presentation of viral and bacterial respiratory infection overlaps, with similar clinical, laboratory and radiological findings hindering the identification of patients who would truly benefit from antibiotic treatment [7,14,15]. Laboratory alterations usually associated with bacterial infections might be prominent in COVID-19, due to a pathologic inflammatory hyperactivation in severe isolate SARS-CoV-2 infection. In the early phases of the pandemic, some decisions had to be made with limited scientific evidence; given the severity of clinical presentation and the unclear course of COVID-19, antibiotics were often prescribed [8,14,16,17]. In outpatient settings, medical practitioners often prescribe antibiotic to prevent co-infection and use macrolides such azithromycin to exploit its immunomodulatory properties and the ability to inhibit viral replication [18,19]. This increase in antibiotic empiric treatment prescription boosts antibiotic resistance in both community and hospital settings [15,20]. Nowadays, no antibiotic is routinely indicated as part of SARS-CoV-2 treatment while immunomodulatory drugs are being increasingly used, hence data on bacterial co-infections and chronic viral infections, such as HIV and hepatitis, assume an important role in inpatient management. Information on the incidence of bacterial co- and super-infections in SARS-CoV-2 infection are scarce and the prevalence of data inconsistent throughout different studies [8,9,11,12,21,22].

The aim of this study is to describe antibiotic prescription practice in COVID-19 patients before hospitalization and bacterial and viral co-infection rates in SARS-CoV-2 infected patients hospitalized during 2021.

## 2. Results

### Study Population

In total, 482 consecutive patients were hospitalized for SARS-CoV-2 infection in the Infectious Diseases ward of the Policlinico Tor Vergata Hospital, Rome, Italy, from 1 January 2021 to 31 December 2021. The median age of the enrolled population was 65 years (interquartile range [IQR] 53–76 years), with a prevalence of males (61.0%) (Table 1). Overall, 75.9% (366/482) of patients required oxygen support during hospitalization: 22.8% needed Venturi oxygen masks, 1.6% non-rebreather oxygen masks with a concentrator, 44.0% non-invasive ventilation and 7.5% mechanical ventilation after orotracheal intubation. The overall mortality was 14.7% (71/482), 30-day mortality was 12.4% (60/482).

The QuantiFERON-TB gold test was performed in 442 patients (91.7%): 9.7% resulted positive, 62% negative and 28.3% indeterminate (Table 2). Serology for hepatitis B virus (HBV) and hepatitis C virus (HCV) was performed for 472 patients (97.9%): 100 patients (21.2%) had positive HBcAb, 6 patients (1.3%) had an active hepatitis B virus infection, with positive HBsAg. Ten patients (2.1%) had positive antibodies for hepatits C virus, 50% were viremic (4 out of 8, viremia not available for 2 patients).

A detailed description of HBV- and HCV-positive patients is available in the Appendix A. A urinary antigen test for legionella pneumophila and streptococcus pneumoniae, was performed in 428 patients (88.8%): 15 patients were positive for *S. pneumoniae*, none for *L. pneumophila* (Table 2). Pharyngeal swabs for respiratory viruses were performed in 389 patients (80.7%) but no respiratory viruses other than SARS-CoV-2 were identified.

Almost half of the patients (237/482, 49.2%) had blood cultures drawn within 48 h of hospital admission; 28 (11.8%) blood cultures were positive, of which only 7 (25%) were considered clinically relevant and treated as bloodstream infection (Table 2). Among the remaining positive blood cultures, which were not considered bloodstream infections due to the presence of contaminating bacteria, 85.7% of the isolated pathogens were skin commensals coagulase-negative staphylococci. No positive blood cultures for fungi were detected. A detailed description of positive blood cultures is available in the Appendix A. During hospitalization, patients with bloodstream infections were treated with a targeted antibiotic therapy, according to antimicrobial susceptibility.

Out of 482 patients, 151 (31.3%) received pre-hospitalization home antibiotic treatment, 9.9% with more than 1 antibiotic. Macrolides were most frequently prescribed (64.9%), followed by oral beta-lactams (22.5%) and fluoroquinolones (2.7%) (Table 2).

Patients were grouped based on the maximal oxygen supply received during hospitalization into non-severe (AA, VRM) and severe (NRM, NIV, OTI). The two subgroups were comparable for age, sex and Charlson comorbidity index (CCI) (Table 3). No differences were reported in the prevalence of viral hepatitis infection; indeterminate results of the QuantiFERON-TB gold test were more frequent in severe than non-severe patients (40.3% vs 15.2%, *p* < 0.001). Rates of positive blood cultures drawn within 48 h from hospital admission were comparable between the two subgroups; home antibiotic treatment was more frequently prescribed to severe patients (38.3% vs. 23.4%, *p* < 0.001).

As for mortality, after stratifying patients into survivors and non-survivors, older age (*p* = < 0.001), higher CCI (*p* < 0.001), cardiovascular (0.003), cerebrovascular (*p* = 0.014), renal (0.012), hematologic comorbidity (0.008) and a solid tumor diagnosis (*p* = 0.001) were associated with higher mortality rates (Table 3). Non-survivors more frequently received a higher oxygen supply (*p* < 0.001), with half of the patients requiring OTI. No differences were reported in the prevalence of viral hepatitis infection; indeterminate QuantiFERON-TB gold test results were more frequent in non-survivors (47.2% vs. 25.7%, *p* = 0.002). Non-survivors more frequently had positive blood cultures at hospital admission (23.5% vs. 9.8%, *p* = 0.021) compared to survivors. After including only positive blood cultures considered as true bloodstream infections, no statistically significant differences were found after comparing the two groups.

## 3. Discussion

In this study we report a low overall prevalence of viral and bacterial co-infections in SARS-CoV-2 hospitalized patients systematically screened at ID ward admission. Nonetheless, home antibiotic treatment was commonly prescribed, mainly to patients with more severe COVID-19 disease.

In known viral respiratory infections, such as influenza, bacterial and fungal co-infection have an important influence on patient outcome [4,5,6,7], therefore in the early stages of the COVID-19 pandemic, antibiotics were commonly prescribed, favored by the scarce pathophysiology knowledge on the newly discovered virus. In Spain, Garcia-Vidal et al. reported that the hospital protocol indicated antibiotic therapy for all hospitalized patients with COVID-19 [11]. As in other viral respiratory diseases, recognition of bacterial co-infections is hampered by the similar clinical and radiological presentation of SARS-CoV-2 infection [7,14,15]. This, coupled with the unsettling encounter of an unknown pathogen with severe clinical manifestation and an uncertain treatment, increased antibiotic use in COVID-19 patients [20].

Later studies have shown that the SARS-CoV-2 virus can predispose to super infections by modifying the respiratory microbiome homeostasis and triggering immune cells to hyper-produce inflammation factors and dysregulate the immune system. It can also damage the respiratory airways and facilitate bacterial adhesion and transmigration [7,23]. SARS-CoV-2 also causes a systemic alteration of both innate and adaptive immune responses, stimulating the decrease in lymphocytes and host immune function; the degree of the virus induced lymphopenia is correlated with more severe COVID-19 disease. This systemic immune modulation together with local lung alteration, might favor super- and secondary infection by bacteria and fungi [24,25]. Despite these profound alterations to the immune system, a relatively low prevalence of bacterial and fungal co-infections has been described in COVID-19 patients, lower than co-infections rates reported for previous respiratory virus pandemics, such as influenza [8,9,11,14,16,22,26,27]. Probably, the infection control and prevention measures adopted by governments worldwide, such as mask wearing, social distancing and lock down strategies contributed to reducing the circulation of other common respiratory infections, like seasonal influenza and pneumococcal disease, which were less frequently diagnosed than in the same periods of previous years [20,28,29].

In our cohort patients were systematically screened for respiratory viruses other than SARS-CoV-2 and none were positive. Data in the published literature are slightly dissimilar, reporting a viral co-infection prevalence of around 3% in SARS-CoV-2 hospitalized patients mainly caused by influenza and respiratory syncytial virus (RSV), with a peak of 20% in the systematic review by Musuuzaa et al. [21], that includes studies on outpatients and children [7,9,11,21,26]. The most commonly reported viruses in this study are influenza A (22.3%), influenza B (3.8%) and RSV (3.8%) [21]. The type and timing of sampling, together with seasonal ecology and infection control measures might influence viral coinfection rates in SARS-CoV-2 patients. In Italy, early stringent lockdown measures were adopted, compared with other countries [16,21,29,30,31].

As for bacterial and fungal co-infection in COVID-19, prevalence varies widely across the published literature, with an overall reported 10–20% co-infection rate [8,9,11,12,14,16,21,22,26,27]. In our cohort we focused on some respiratory co-infections, screening hospitalized patients for respiratory viruses other than SARS-CoV-2 and urinary antigens for *Legionella pneumophila* and *Streptococcus pneumoniae.* Sputum samples were not systematically collected due to technical difficulties, hence even if available, sputum results were not included in the analysis. This represents a limitation of the study, considering that atypical respiratory pathogens were not thoroughly investigated. *S. pneumoniae* was the most detected cause of co-infection (15 patients, 3.5%), consistent with the published literature [21]. A recent retrospective study conducted in Spain by Moreno-Garcia E et al., analyzed co-infection in 1125 SARS-CoV-2 patients hospitalized between February 2020 and February 2021, collecting blood cultures, urinary antigens for *S. pneumoniae* and *L. Pneumophila* and respiratory samples, reported a similar prevalence of co-infection (9.1%), mainly due to *S. pneumoniae* (79%), *S. aureus* (6.8%) and *H. influenzae* (6.8%) [32]. A smaller retrospective study by Rothe et al. [15] evaluated co-infections and superinfections in 140 hospitalized SARS-CoV-2 patients, with a study design and sample collection strategy similar to our study, detected 0 positive urinary antigen tests for *L. pneumophila* and *S. pneumoniae*, and 7.1% positive blood cultures, with 4.2% clinically relevant bacteremia.

A review by Lars F. Westblade et al. [8] reported S. aureus (31%), *S. pneumoniae* (23%) and *H. influenzae* as the most common pathogens from blood and respiratory samples, and a rate of positive blood cultures at hospital admission of around 1.2–4.2% of cases, with almost half of the isolates being skin contaminants. In our cohort, blood cultures were collected in 237 patients within 48 h of hospital admission, with a positive rate of 11.8%, and 2.9% were considered as clinically relevant bacteremia; contaminants were detected in 75% of cases. This high prevalence of contaminants in blood samples might be due to technical challenges in collecting blood cultures with personal protective devices, further complicated by the emergency setting in which COVID-19 hospitals operated, with bed and laboratory capacity intensely strained [8]. Given the high rate of contaminations, probably blood cultures should be drawn only in hospitalized patients with clinically suspected bacteremia, to optimize resource utilization and increase diagnostic yield [8,15,20].

Antimicrobial treatment, when needed, might be focused towards the most common pathogens, such as *S. pneumoniae* and *S. aureus*, following local epidemiology for drug susceptibilities. Despite the low bacterial co-infection rate at hospital admission, 31.3% of the patients in our cohort received pre-hospital home antibiotic therapy. This is probably correlated with the perceived uncertainty of treatment for a new disease and the known immunomodulatory properties of macrolide antibiotics [11,18,19,33]. Also in hospital settings, empirical antibiotic therapy has been widely overprescribed in SARS-CoV-2 patients, as reported in the systematic review by Musuuza et al., where 98% of the included papers reported antibiotic prescription in COVID-19 hospitalized patients [21]. In a large randomized cohort of 1705 hospitalized COVID-19 patients in the United States of America [22], antibiotics were prescribed to 56.6% of the population, despite an overall prevalence of 3.5% of community onset bacterial co-infections. Patients were more likely to receive empiric antibacterial therapy if they were older, had a lower body mass index, had more severe comorbidity (e.g., respiratory support, severe sepsis), had a lobar infiltrate, or higher inflammatory biomarkers [22]. When the clinical and radiological features of different diseases overlap, as in SARS-CoV-2 and bacterial respiratory co-infections, diagnostic stewardship needs to be improved, especially in limited resource settings. Biomarkers such as procalcitonin might be useful and proper microbiologic sampling, in both time and type, assumes a pivotal role in optimizing patient care [8,14,21,26].

Macrolides were the antimicrobials more frequently prescribed in our cohort (64.9%), and in some settings their use was recommended early on in SARS-CoV-2 patients, due to their immunomodulatory properties [17,18,22,33]. In the latest guidelines on COVID-19 treatment released in 2022 by the European Society of Infectious Diseases (ESCMID), no beneficial effect of azithromycin compared with standard of care was demonstrated, while it showed severe side effects, such as a prolonged corrected QT interval [34]. Given the observed low rate of bacterial coinfections, antibiotics should not be routinely prescribed in patients with COVID- 19, unless a bacterial coinfection or secondary infection is suspected or confirmed [34]. Unnecessary use of antimicrobial agents is associated not only with a significant economic burden on healthcare systems [20], but also with increased side-effects for patients and an increase in multi-drug resistance [15,20], as described in 2003 during the spread of SARS-CoV [15]. Guidelines and clinicians should focus on adequate microbiologic sampling strategies prior to antibiotic administration and the definition of targeted antimicrobial therapy, towards the more common community-acquired pathogens, to limit antibiotic overuse and fight emerging resistance, even in the context of a pandemic [15,35]. Once detected, bacterial and fungal co-infections in SARS-CoV-2 infected patients should be promptly treated, given the higher associated mortality described in the published literature [9,21]. In our cohort, positive blood cultures were statistically associated with higher mortality, but when we consider blood cultures treated as infection and blood cultures considered as contaminants, there was no association with the outcome.

Patients with more severe diseases received home antibiotic treatment more often, even if the prevalence of bacterial co-infection was not significantly different from non-severe patients. Interestingly, patients prescribed home antibiotics seem to present later to the hospital (6.1 vs. 7.9 days, *p* < 0.001), probably due to the placebo effect of receiving a drug, even with unknown clinical benefits. Nevertheless, higher home antibiotic prescription in severe COVID-19 patients might also account for a more severe infection presentation from the beginning, triggering both doctors and patients to promptly start assuming drugs. Overall, no beneficial effect seems to derive from early antibiotic prescription in SARS-CoV-2 infection without bacterial co-infection.

The QuantiFERON-TB gold test was performed in 442 patients (91.7%), with an overall positive rate of 1.9% and a statistically significant higher prevalence of indeterminate results in both severe patients and non-survivors, as previously shown by our research group [36,37,38]. The high prevalence of indeterminate QFT-Plus assay in a large cohort of patients hospitalized because of SARS-CoV-2 infection, is directly linked to the impaired IFN-γ production in the mitogen-nil condition and the reduction in peripheral blood T-lymphocytes in COVID-19 patients.

There are limited data on chronic viral hepatitis screening in hospitalized COVID-19 patients. In our cohort, 97.9% of patients were screened, with detection of 3 HCV-related chronic hepatitis (0.6%), 6 HBV-related chronic infections (1.3%) and 100 HBcAb positive subjects (21.2%). These seroprevalence rates are similar to the Italian epidemiology for both HBV and HCV [39]. Chronic viral hepatitis does not seem to have a direct impact on COVID-19 outcome, as reported in a systematic review by Srakar et al. [40], that shows comparable mortality rates in patients with and without chronic viral hepatitis and SARS-CoV-2 infection, while cirrhosis increases the risk of mortality [41]. Particular attention should be given to hepatitis B screening and co-infection diagnosis, considering the profound impairment of the immune system of severe COVID-19 patients. Immune impairment might also be increased by immunosuppressive therapies used for COVID-19, such as steroids and interleukin inhibitors. As other immunosuppressed subjects, HBsAg positive as well as HBsAg negative HBcAb COVID-19 positive patients face the risk of HBV reactivation. A recently published review suggest HBV screening for all hospitalized SARS-CoV-2 patients, with HBV-DNA quantification also in HBsAg negative HBcAb positive subjects to properly assess reactivation risk and start the appropriate antiviral prophylaxis, if needed, during iatrogenic-induced immunosuppression [42]. In patients with chronic, occult or resolved HBV and SARS-CoV-2 infection receiving high doses of steroids or other immunosuppressive agents, antiviral therapy for HBV might be considered to prevent viral flares or reactivation [41].

The present study has some limitations, namely its retrospective and monocentric design, making the findings difficult to generalize. It also has some strengths, with a large number of patients included who underwent a systematic microbiologic screening.

## 4. Materials and Methods

### 4.1. Study Design and Participants

The study is a single-center, retrospective, observational study, performed at the Policlinico Tor Vergata University Hospital of Rome, Italy. All adults (≥18 years), hospitalized for SARS-CoV-2 in the Infectious Diseases Clinic of the Policlinico Tor Vergata University Hospital, from 1 January 2021 to 31 December 2021, with a positive reverse transcription polymerase chain reaction (RT-PCR) for SARS-CoV-2 on a nasopharyngeal (NPh) swab were included. The study was approved by the local ethics committee (experimentation register number 14.22). Given the retrospective nature of the study, written informed consent was not necessary. The study was conducted in accordance with the principles of the Declaration of Helsinki.

### 4.2. Data Collection

An ad hoc electronic database was created to collect clinical data, including demographic data, comorbidities, laboratory and microbiology data, oxygen support and ventilation type (either non-invasive or invasive), outcome (in-hospital mortality). Clinical data were directly registered from clinical charts; laboratory and microbiology data were extracted from the electronic hospital software (Modulab version 3.1.02). All blood tests were performed in the central laboratory of the Policlinico Tor Vergata University Hospital, following standard procedures.

### 4.3. Laboratory Analyses

The GeneFinderTM COVID-19 Plus RealAmp Kit, ELITech AllplexTM 2019-nCoV Assay (Seegene, Seoul, South Corea) was used for real time-PCR. It is based on the identification of three genetic viral targets of SARS-CoV-2: Envelope (E), Nucleocapsid (N) and RNA-dependent RNA-Polymerase (RdRP) genes.

At the infectious diseases ward admitted patients were screened for:-Human immunodeficiency virus (HIV) (Alinity HIV, Abbott Molecular, Des Plaines, IL, USA);-Hepatitis C antibodies (Alinity HCV, Abbott Molecular, Des Plaines, IL, USA);-Hepatitis B antibodies (HBcAb, HBSAb, HBSAg, HBeAb, HBeAg) (Alinity HBV, Abbott Molecular, Des Plaines, IL, USA);-Urinary antigens for *Legionella pneumophila* and *Streptococcus pneumoniae* (Sofia Legionella FIA Quidel, Sofia S.pneumoniae FIA Quidel, San Diego, CA, USA);-Pharyngeal swab for respiratory viruses (Biofire Filmarray Respiratory 2.1 plus Panel, BioFire Diagnostics, LLC, Salt Lake City, UT, USA; it detects viral PCR for respiratory sincitial virus, metapneumovirus, coronaviruses, influenza virus, parainfluenza virus, rhinovirus/enterovirus);-QuantiFERON^®^-TB Gold Plus (QFT-Plus) assay (Qiagen, Hilden, Germany).

If performed according to a physician’s clinical decision, blood cultures collected within 48 h of hospital admission (considering the emergency department arrival) were recorded. Blood cultures were collected following the manufacturer’s instructions and standard laboratory procedures for pathogen identification and antimicrobial susceptibility testing.

### 4.4. Definitions: Patient Classification

Patients were stratified into 5 groups according to the maximal oxygen supply/ventilation support required during the hospitalization: ambient air (AA), Venturi oxygen mask (VMK), non-rebreather oxygen mask with concentrator (NRM), non-invasive ventilation (NIV), and invasive mechanical ventilation through orotracheal intubation (OTI). These subpopulations were further classified as non-severe, patients in the AA and VMK groups, and severe, patients in the NRM, NIV and OTI groups. Patients were defined as non-survivors if they died within 30-days after hospitalization and survivors if they: (1) were discharged at home, (2) remained in hospital or were moved to a residential structure for COVID-19 patients because of public health issues, (3) were moved to another hospital and were still alive after 30 days from first hospitalization.

### 4.5. Statistical Analysis

Continuous data are presented as median with interquartile range (IQR), while categorical data are presented as absolute frequencies with percentages. Differences between groups were assessed using the Mann–Whitney U test (two groups, continuous variable), the Kruskal–Wallis test (more than two groups, continuous variable) or the Chi^2^ test (categorical variables). Statistical analyses were performed using the software JASP (version 0.17.0 JASP Team, 2019).

## 5. Conclusions

In our cohort, bacterial community-acquired co-infections in SARS-CoV-2 infected hospitalized patients were rare, and did not seem to influence COVID-19 severity and outcome. Nonetheless, antibiotics were frequently prescribed before hospitalization, highlighting the need to improve diagnostic and antimicrobial stewardship.

Chronic viral hepatitis screening should be recommended in patients hospitalized for SARS-CoV-2, especially in high prevalence areas, to safely manage immune modulating treatment in COVID-19 patients.

## Figures and Tables

**Table 1 antibiotics-12-01348-t001:** Overall population clinical characteristics.

Overall Population 482 Patients	
Age (median [IQR])	65 (53–76)
Sex (M/F)	294/188 (61%/39%)
Time to hospitalization from symptom onset (median [IQR])	6 (3–9)
CCI (median (IQR))	4 (2–5)
Comorbidities	
Cardiovascular	284 (58%)
Diabetes	101 (21%)
Obesity	97 (20.1%)
Psychiatric/Neurologic	64 (13.3%)
Pulmonary	61 (12.6%)
Endocrinologic	61 (12.6%)
Renal	57 (11.8%)
Solid Tumor	54 (11.2%)
Cerebrovascular	35 (7.3%)
Hematologic	28 (5.8%)
Immuno/Rheumatologic	28 (5.8%)
Hepatitis	25 (5.5%)
Dialysis	20 (4.1%)
Other	170 (35.3%)
Oxygen Supply	366 (75.9%)
AA/VMK/NRM/NIV/OTI	116/110/8/212/36 (24.1%/22.8%/1.6%/44%/7.5%)
Overall mortality	71 (14.7%)
30-day mortality	60 (12.4%)

IQR: interquartile range; CCI: Charlson comorbidity index; AA: ambient air; VMK: Venturi oxygen mask; NRM: non-rebreather oxygen mask with concentrator; NIV: non-invasive ventilation; OTI: invasive mechanical ventilation through orotracheal intubation.

**Table 2 antibiotics-12-01348-t002:** Overall population co-infections.

**Overall Population** **482 Patients**	
QuantiFERON-TB Gold	442 pts
Positive/indeterminate/negative	43/125/274 (9.7%/28.3%/62%)
Hepatitis screening	472 pts
HBcAb+	100 (21.2%)
HbsAg+	6 (1.3%)
Anti-HCV+	10 (2.1%)
Urinary antigen	482 pts
*Legionella pneumophila*	0
*Streptococcus pneumoniae*	15 (3.5%)
Pharyngeal swab–respiratory viruses	389 pts
Positive swabs	0
Blood cultures within 48 h from hospitalization	237 (49.2%)
Positive/negative	28/209 (11.8%/88.2%)
Contaminant/Infection	21/7 (75%/25%)
Home antibiotic treatment	151 (31.3%)
With 1 antibiotic	136 (90.1%)
macrolides	98 (64.9%)
oral beta-lactams	34 (22.5%)
fluoroquinolones	4 (2.7%)
With more than 1 antibiotic	15 (9.9%)

pts: number of patients tested; HBcAb: antibodies anti-hepatitis B core; HBsAg: hepatitis B surface antigen; Anti-HCV+: Anti-HCV antibody positivity.

**Table 3 antibiotics-12-01348-t003:** Clinical characteristics and co-infection screening in severe and non-severe patients and in survivors and 30-day non-survivors.

	Non-Severe 226 Patients (46.9%)	Severe 256 Patients (53.1%)	*p*-Value	Survivors 422 Patients (87.6%)	Non-Survivors 60 Patients (12.4%)	*p*-Value
Age (median (IQR))	64 (51–77)	65 (55–74)	0.745	63 (51–74)	77 (69–82.2)	**<0.001**
Sex (M/F)	136/90 (60.2%/39.8%)	158/98 (61.7%/38.3%)	0.729	260/162 (38.4%/61.6%)	34/26 (56.7%/43.3%)	0.462
Time to hospitalization from symptom onset (median (IQR))	6 (3–8.5)	7 (4–10)	0.100	5.5 (3–9)	7 (4–9)	0.261
CCI (median (IQR))	4 (1–5)	3.5 (2–5)	0.501	3 (1–5)	5 (4–7)	**<0.001**
Comorbidities						
Cardiovascular	135 (59.7%)	149 (58.2%)	0.733	238 (56.4%)	46 (76.7%)	**0.003**
Diabetes	44 (19.5%)	57 (22.3%)	0.452	85 (20.1%)	16 (26.7%)	0.245
Obesity	33 (14.6%)	64 (25%)	**0.004**	82 (19.4%)	15 (25%)	0.314
Psychiatric/Neurologic	28 (12.4%)	36 (14.1%)	0.589	54 (12.8%)	10 (16.7%)	0.408
Pulmonary	28 (12.4%)	33 (12.9%)	0.869	51 (12.1%)	10 (16.7%)	0.318
Endocrinologic	23 (10.2%)	38 (14.8%)	0.124	53 (12.6%)	8 (13.3%)	0.866
Renal	35 (15.5%)	22 (8.6%)	**0.019**	44 (10.4%)	13 (21.7%)	**0.012**
Solid Tumor	21 (9.3%)	33 (12.9%)	0.211	40 (9.5%)	14 (23.3%)	**0.001**
Cerebrovascular	20 (8.8%)	15 (5.8%)	0.207	26 (6.2%)	9 (15%)	**0.014**
Hematologic	11 (4.9%)	17 (6.6%)	0.406	20 (4.7%)	8 (13.3%)	**0.008**
Immuno/Rheumatologic	14 (6.2%)	14 (5.5%)	0.734	25 (5.9%)	3 (5%)	0.775
Hepatitis	14 (6.2%)	11 (4.3%)	0.348	24 (5.7%)	1 (1.7%)	0.189
Dialysis	17 (7.5%)	3 (1.2%)	**<0.001**	16 (3.8%)	4 (6.7%)	0.296
Other	82 (36.3%)	88 (34.4%)	0.662	155 (36.7%)	15 (25%)	0.075
Oxygen Supply (AA/VMK/NRM/NIV/OTI)	//	//	//	113/105/4/194/6 (26.8/24.9/0.9/46/1.4)	3/5/4/18/30 (5/8.3/6.7/30/50)	**<0.001**
QuantiFERON-TB Gold (positive/indeterminate/negative)	28/32/151 (13.3/15.2/71.6)	15/93/123 (6.5/40.3/53.2)	**<0.001**	42/100/247 (10.8/25.7/63.5)	1/25/27 (1.9/47.2/50.9)	**0.002**
HBcAb+	51 (23.2%)	49 (19%)	0.322	85 (20.5%)	15 (25.9%)	0.352
HbsAg+	1 (0.4%)	5 (2%)	0.139	6 (1.4%)	0	0.356
Anti-HCV+	6 (2.7%)	4 (1.6%)	0.391	10 (2.4%)	0	0.232
Urinary antigen (*L. pneumophila*/*S. pneumoniae*)	0/10 (0%/5.1%)	0/5 (0%/2.1%)	0.091	0/15 (0%/4%)	0/0	0.143
Pharyngeal swab–respiratory viruses	0	0		0	0	
Blood cultures within 48 h from hospitalization	106	132		204	34	
Positive/negative	11/95 (10.4%/89.6%)	17/115 (12.9%/87.1%)	0.552	20/184 (9.8%/90.2%)	8/26 (23.5%/76.5%)	**0.021**
Contaminant/Infection	7/4 (63.6%/36.4%)	14/3 (82.3%/17.7%)	0.264	14/6 (70%/30%)	7/1 (87.5%/12.5%)	0.334
Home antibiotic treatment	53 (23.4%)	98 (38.3%)	**<0.001**	136 (32.2%)	15 (25%)	0.259

IQR: Interquartile range; M: male; F: female; AA: ambient air; VMK: Venturi oxygen mask; NRM: non-rebreather oxygen mask with concentrator; NIV: non-invasive ventilation; OTI: invasive mechanical ventilation through orotracheal intubation; HBcAb: antibodies anti-hepatitis B core; HBsAg: Hepatitis B surface antigen; Anti-HCV+: Anti-HCV antibody positivity. The bolded values represent statistically significant parameters.

## Data Availability

The data presented in this study are available on request from the corresponding author.

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
