# Peer review of "The Impact of Viral and Bacterial Co-Infections and Home Antibiotic Treatment in SARS-CoV-2 Hospitalized Patients at the Policlinico Tor Vergata Hospital, Rome, Italy"

_antibiotics, 2023, doi:10.3390/antibiotics12091348_

Round 1

Reviewer 1 Report

Lorenzo et al.’s article titled The impact of viral and bacterial co-infections and home antibiotic treatment in SARS- CoV-2 hospitalized patients at Policlinico Tor Vergata Hospital, Rome, Italy. The starting point of this article is commendable, the authors aimed to assess the presence of co-infections among patients hospitalized for SARS-CoV-2 infection,but there are still some issues as indicated below:

1.     The article mentions the relationship between co-infections and clinical outcomes in patients. To enhance credibility, consider using methods like regression analysis to explore potential associations between co-infections and factors such as oxygen status, mortality rate and length of hospital stay.

2.     The study design of this article is narrow, and the conclusions drawn may not be comprehensive. For example, the scope of co-infections includes viral, bacterial, fungal, and atypical pathogens. The authors should discuss the reasons for not detecting fungi and atypical pathogens in the article.

3.     The article mentions the possibility of antibiotic overuse in COVID-19 patients. How does the conclusion of this study impact this phenomenon?

4.     The article mentions certain methodological limitations, such as challenges in sample collection leading to the inability to collect sputum samples. In the discussion section, a more in-depth exploration of these limitations' impact on the study results and their potential effects on the detection and assessment of co-infections can be provided.

5.     The conclusion section is too lengthy. It is recommended to simplify and condense it for clarity and ease of reading.

Moderate editing of English language required

Author Response

We thank the Reviewer for the comments and suggestions.

  1. The article mentions the relationship between co-infections and clinical outcomes in patients. To enhance credibility, consider using methods like regression analysis to explore potential associations between co-infections and factors such as oxygen status, mortality rate and length of hospital stay.

We evaluated the potential association between co-infections, outcomes and disease severity.

Concerning the outcomes, after performing a multiple logistic regression analysis, which including in the model 1) blood cultures within 48 hours from hospitalization; 2) QuantiFERON-TB Gold; 3) HbsAg+; 4) HCV+; 5); urinary antigen detection, no statistically significant associations were identified. However, an indeterminate result at the QuantiFERON-TB Gold assay and positive blood cultures within 48 hours from hospitalization were associated to a reduced 30-day survival, at the limit of the statistical significance (odds ratio: 0.150, p=0.08 and 0.359, p=0.06, respectively).

Concerning disease severity, which in our paper corresponds to patients’ Oxygen status (severe patients were those needing non-invasive or invasive ventilation or non-rebreather oxygen mask with concentrator, while non-severe patients were those not needing any Oxigen supply or only Venturi mask), after performing a multiple logistic regression analysis, which including in the model 1) blood cultures within 48 hours from hospitalization; 2) QuantiFERON-TB Gold; 3) HbsAg+; 4) HCV+; 5); urinary antigen detection, no statistically significant associations were identified. However, an indeterminate result at the QuantiFERON-TB Gold assay was associated to an increased severity, at the limit of the statistical significance (odds ratio: 2.788, p=0.07).     

Unfortunately, we did not record the length of hospital stay in our database, as we considered as outcomes the 30-day mortality and clinical severity during hospitalization.

The logistic regression analysis confirmed the results obtained with the univariate analysis as described in the result section of the paper, lines 106-133.

  1. The study design of this article is narrow, and the conclusions drawn may not be comprehensive. For example, the scope of co-infections includes viral, bacterial, fungal, and atypical pathogens. The authors should discuss the reasons for not detecting fungi and atypical pathogens in the article.

The scope of the study was to investigate the antibiotic prescription practice in COVID-19 patients before hospitalization and bacterial and viral co-infections rate in SARS-CoV-2 infected patients hospitalized during 2021, in a real life setting.

We agree with the reviewer when he/she says that fungal and atypical pathogens were not thoroughly investigated in our paper, thus representing a limitation of the study. Fungi were only investigated through traditional cultures on blood samples without any positivity (line 98).

Respiratory tract samples were obtained in a very limited number of patients; consequently, they were not included in the analysis. Atypical pathogens such as Mycoplasma and Chlamydia were not systematically investigated.  Conversely Legionella was systematically assessed through urinary antigen detection. We have already pointed out this aspect in the text: lines 183-185.

To further clarify this aspect we added the following sentence (lines 185-186):

This represents a limitation of the study, considering that atypical respiratory pathogens were not thoroughly investigated.

  1. The article mentions the possibility of antibiotic overuse in COVID-19 patients. How does the conclusion of this study impact this phenomenon?

The results of our study highlight the very limited utility of antibiotic use in COVID-19 patients before hospital admission. Being a real life study, we believe that these results can help clinicians to focus on limiting antibiotic overuse and fight emerging resistance also in a pandemics context.

Conclusions and related impact are described in lines 241-250.

  1. The article mentions certain methodological limitations, such as challenges in sample collection leading to the inability to collect sputum samples. In the discussion section, a more in-depth exploration of these limitations' impact on the study results and their potential effects on the detection and assessment of co-infections can be provided.

As described in the answer to question n. 2, a sentence reporting the limited possibility to detect atypical pathogens in sputum samples was added in lines 185-186.

  1. The conclusion section is too lengthy. It is recommended to simplify and condense it for clarity and ease of reading.

The Discussion section and Conclusions have been revised and simplified.

Reviewer 2 Report

This retrospective study presents a well-articulated analysis of various factors influencing the survival rate of COVID-19-infected patients.

Upon reviewing this paper, I'd like to offer two observations:

The administration of antibiotics at home before hospitalization could potentially elevate the false negative rate in blood cultures. This consequence may lead to an erroneous negative correlation between coinfection and SARS-CoV-2 mortality. It would be valuable to ascertain the prevalence of prior antibiotic treatment among both survivors and non-survivors. This could shed light on the extent of this phenomenon.

The author highlights a noteworthy observation that a substantial proportion of positive culture results are, in fact, false positives with no clinical relevance. Reevaluating the relationship between coinfection and mortality by excluding these false positive cases could yield a more accurate understanding of the association. Could you provide insights into the altered outcomes after this exclusion?

Author Response

Upon reviewing this paper, I'd like to offer two observations:

The administration of antibiotics at home before hospitalization could potentially elevate the false negative rate in blood cultures. This consequence may lead to an erroneous negative correlation between coinfection and SARS-CoV-2 mortality. It would be valuable to ascertain the prevalence of prior antibiotic treatment among both survivors and non-survivors. This could shed light on the extent of this phenomenon.

We thank the reviewer for the observation. As reported in table 3, prior antibiotic treatment was not associated to increased mortality. Indeed, 32% of survivors vs 25% of non-survivors received an antibiotic treatment before hospitalization, without any statistically significant difference (p=0.259). Conversely more severe patients seemed to be more frequently treated with antibiotics before hospital admission (23.4% of non-severe vs 38.3% of severe patients, p<0.001).   

The author highlights a noteworthy observation that a substantial proportion of positive culture results are, in fact, false positives with no clinical relevance. Reevaluating the relationship between coinfection and mortality by excluding these false positive cases could yield a more accurate understanding of the association. Could you provide insights into the altered outcomes after this exclusion?

We thank the reviewer for the observation. We performed a statistical analysis considering only bloodstream infections, after excluding false positive blood cultures. No statistically significant difference was detected, although the number of infections was very low: 7 total bloodstream infections (1 in the non-survivors group and 6 in the survivors group).

We added a sentence in the result section to outline this point (lines 131-133):

“After including only positive blood cultures considered as true bloodstream infections, no statistically significant differences were found after comparing the two groups.”

Reviewer 3 Report

Please add comments on outcomes, especially of those patients with bacteremia.

What antibiotics were administered before admission?

What treatment is received by patients with positive blood cultures?

Author Response

Please add comments on outcomes, especially of those patients with bacteremia.

We thank the reviewer for the observation.

We added a sentence in the result section to clarify this point (lines 131-133):

“After including only positive blood cultures considered as true bloodstream infections, no statistically significant differences were found after comparing the two groups.”

What antibiotics were administered before admission?

We thank the reviewer for the comment. Number and types of antibiotics administered before hospital admission are reported in table 2.

What treatment is received by patients with positive blood cultures?

6/7 bloodstream infections were due to GRAM positive bacteria. Antibiotic treatment was targeted to the microbial isolate also considering the antimicrobial susceptibility tests.

Round 2

Reviewer 1 Report

The author has made revisions in accordance with the review comments to the best of their ability and agrees to publish in the current form.